# IRF1 Mediates Growth Arrest and the Induction of a Secretory Phenotype in Alveolar Epithelial Cells in Response to Inflammatory Cytokines IFNγ/TNFα

**DOI:** 10.3390/ijms25063463

**Published:** 2024-03-19

**Authors:** Giulia Recchia Luciani, Amelia Barilli, Rossana Visigalli, Roberto Sala, Valeria Dall’Asta, Bianca Maria Rotoli

**Affiliations:** Laboratory of General Pathology, Department of Medicine and Surgery, University of Parma, 43125 Parma, Italyroberto.sala@unipr.it (R.S.); biancamaria.rotoli@unipr.it (B.M.R.)

**Keywords:** cell cycle arrest, IFNγ, IRF1, senescence, SASP, TNFα

## Abstract

In COVID-19, cytokine release syndrome can cause severe lung tissue damage leading to acute respiratory distress syndrome (ARDS). Here, we address the effects of IFNγ, TNFα, IL-1β and IL-6 on the growth arrest of alveolar A549 cells, focusing on the role of the IFN regulatory factor 1 (IRF1) transcription factor. The efficacy of JAK1/2 inhibitor baricitinib has also been tested. A549 WT and IRF1 KO cells were exposed to cytokines for up to 72 h. Cell proliferation and death were evaluated with the resazurin assay, analysis of cell cycle and cycle-regulator proteins, LDH release and Annexin-V positivity; the induction of senescence and senescence-associated secretory phenotype (SASP) was evaluated through β-galactosidase staining and the quantitation of secreted inflammatory mediators. While IL-1 and IL-6 proved ineffective, IFNγ plus TNFα caused a proliferative arrest in A549 WT cells with alterations in cell morphology, along with the acquisition of a secretory phenotype. These effects were STAT and IRF1-dependent since they were prevented by baricitinib and much less evident in IRF1 KO than in WT cells. In alveolar cells, STATs/IRF1 axis is required for cytokine-induced proliferative arrest and the induction of a secretory phenotype. Hence, baricitininb is a promising therapeutic strategy for the attenuation of senescence-associated inflammation.

## 1. Introduction

The role of cytokines as a component or trigger of the inflammatory response in lung pathologies is, nowadays, widely recognized. Besides regulating both the initiation and maintenance of immune and inflammatory responses, cytokines also induce and modulate cell proliferation, differentiation, movement and death. As a consequence, a dynamic balance of cytokine levels is strictly required, and the release of pro- and anti-inflammatory elements is necessary to eliminate the pathogen and limit the host damage [1].

The perturbation of this equilibrium can lead to cytokine storm syndrome (CSS), in which an hyperactivation of the innate immune system causes an uncontrolled and prolonged release of pro-inflammatory signaling molecules that causes tissue damage, multisystem organ failure and death [2].

In the lung, in particular, cytokine storm can result in excessive immune cell infiltrations in the pulmonary tissues, leading to severe lung damage. Consistently, the excessive presence of inflammatory mediators is central to the pathogenesis of acute respiratory distress syndrome (ARDS) and acute lung injury (ALI), deleterious conditions characterized by alveolar epithelial and lung endothelial injury leading to pulmonary edema, acute ipoxemia and accumulation of bilateral pulmonary infiltrates [3]. ALI/ARDS can be secondary to a wide range of diseases such as pneumonia, sepsis and traumas. The pathophysiology of ARDS is complex; in the early phase, mechanisms for pulmonary edema formation include dysfunction of the alveolar–capillary barrier, recruitment of polymorphonuclears and an increased amount of cytokines in plasma and bronchoalveolar lavage fluid, mainly Interleukin-1β (IL-1β), Interleukin-6 (IL-6), Interleukin-8 (IL-8) and Tumor Necrosis Factor-α (TNFα) [4]. In the later stage, the injured lung is characterized by severe fibrosis and alveolar destruction and reconstruction. A recent meta-analysis study, aimed to correlate ARDS/ALI with inflammatory factors, identified an association between this pathology and increased levels of Angiopoietin 2 (ANG-2), IL-1β, IL-6 and TNFα, but not with IL-8, Interleukin-10 (IL-10) and Plasminogen activator inhibitor-1 (PAI-1) levels [5].

Also, COVID-19 is an important cause of ARDS, which is even the most predictive factor for death in critically ill patients [6]. In COVID-19 patients, the level of inflammatory cytokines is increased and plays a crucial role in the pathogenesis of the disease; among them, interferon-γ (IFNγ), TNFα, IL-1β and IL-6 are indicated as the key cytokines associated with the severity of COVID-19 [7,8,9].

IL-1β is a pleiotropic cytokine mainly produced by macrophages that, similarly to TNFα, has a major impact on cell proliferation, differentiation and cell death. It is one of the cytokines most commonly found in pulmonary edema and bronchoalveolar lavage fluids in ARDS [10,11]. In pulmonary inflammation, IL-1β increases lung barrier permeability in in vitro and in vivo models of ARDS and may contribute to alveolar edema in lung injury models by impairing fluid reabsorption from the lungs [12].

IL-6 is a glycoprotein expressed by different cell types, including lymphocytes, monocytes/macrophages, dendritic cells and endothelial cells. It is an established biomarker of inflammation with low plasma concentrations under normal physiological conditions that rise under various pathological conditions [13,14]. IL-6 has been demonstrated to play a leading role in cytokine storm syndrome and, as for COVID-19, the severity of the disease positively correlates with enhanced IL-6 and C-reactive protein serum levels [15].

TNFα is a classical cytokine produced upon local or systemic inflammation, regulating differential processes such as cell proliferation and differentiation as well as cell death [16]. In the alveolar epithelium, the TNFα/TNFR1 interaction, besides directly regulating the activity of ion channels and pumps, also modulates the integrity of the alveolar barrier [17]. It can also directly deteriorate the respiratory epithelium by inducing the production of other inflammatory mediators such as IL-8, granulocyte macrophage-colony stimulating factor (GM-CSF) and of intercellular adhesion molecules (ICAMs) [18].

IFNγ plays a central role in the antiviral and antitumor responses by orchestrating both innate and adaptive immunity [19,20]. IFNγ is endowed with many biological functions, including cytostatic, proapoptotic and antiproliferative effects [21,22]. The binding of the cytokine to its receptor (IFNγR) results in the activation of the JAK1/2-STAT1 pathway that, through interferon regulatory factor 1 (IRF1), promotes the downstream transcriptional network [23,24]. Regarding SARS-CoV-2 infection, the exact role of IFNγ appears complex and contradictory findings are reported [25]. Indeed, although IFNγ is able to inhibit the SARS-CoV-2 replication [26,27], elevated levels of IFNγ, especially in combination with TNFα, aggravates the systemic inflammation causing tissue injury and organ failure [28,29]. On the other hand, it is also reported that the expression of IFNγ is significantly reduced in severe COVID-19 hospitalized patients [30].

These four cytokines have also been included in the group of mediators released by senescent and tumor cells in the complex pro-inflammatory response known as the senescence-associated secretory phenotype (SASP) [31]. Hallmarks of senescence, besides stable cell cycle arrest, are, indeed, the secretion of inflammatory cytokines and chemokines, ECM-degrading signals, as well as the upregulation of cell surface molecules [32].

Recently, by addressing the effect of cytokines released by human macrophages treated with SARS-CoV-2 spike S1 protein on A549 alveolar epithelial cells, we observed that A549 cells undergo a cell cycle arrest [33]. In order to identify which of the many cytokines released by S1-activated macrophages is responsible for growth arrest, we evaluate here the effect of IFNγ, TNFα, IL-1β and IL-6 on the proliferation of A549 cells, mainly focusing on the role of IRF1 transcription factor.

## 2. Results

### Effects of Cytokines on Cell Viability in A549 WT and IRF1 KO A549 Cells

The effect of IFNγ, TNFα, IL-1β and IL-6 on cell viability has been preliminary evaluated in WT and IRF KO A549 cells upon incubation for 72 h with increasing concentrations (from 0.1 to 50 ng/mL) of each cytokine (Figure 1A). While IL-1 and IL-6 had no effect on cell viability, a modest but significant dose-dependent decrease has been observed in both cell models upon treatment with TNFα. The effect of IFNγ was, instead, specific for WT and IRF1 KO cells; in WT, a marked decrease in cell viability was observed even with 0.1 ng/mL of the cytokine, although the maximal inhibition (about 60%) was detectable at 10 ng/mL. Under the same experimental conditions, instead, only a modest (about 15%) reduction in cell viability was observed in IRF KO cells.

The cooperative effects of the cytokines were, then, addressed. To this end, A549 cells were incubated for 72 h with combinations of the different cytokines (Figure 1B). In light of the results shown in Panel A, all cytokines have been employed at the concentration of 50 ng/mL, with the exception of IFNγ that was used at 10 ng/mL, i.e., the minimal concentration causing the maximal inhibition of cell viability. Data obtained indicate that the reduction in cell viability observed in WT cells upon incubation with IFNγ was further strengthened by the simultaneous presence of TNFα, when the cell population roughly decreased to 20% of control untreated cells. Interestingly, the same combination of cytokines, namely IFNγ + TNFα, exerted only a modest, although significant, decrease in cell viability in IRF1 KO cells. The presence or addition of IL-1β and IL-6 had no effect on the viability of either cell model, excluding a role for these mediators in the modulation of cell viability in A549 epithelial cells.

A time-course analysis of cell viability was then performed in A549 WT and IRF1 KO cells upon exposure to IFNγ + TNFα (cytomix) for up to 72 h. As shown in Figure 2A, cell proliferation was comparable in both cell models under control conditions. The incubation with cytomix caused, instead, a complete arrest of cell growth in WT cells, but not in IRF1 KO, where a slight significant reduction in cell population was observed only after 72 h. At this time, cell morphology was deeply different, with WT cells looking sparse and with an elongated shape, while IRF KO cells had no appreciable modification with respect to control untreated cultures. No sign of cell detachment or death was evident under any experimental condition, suggesting a cytostatic, rather than a cytotoxic effect of cytomix in A549 cells. Consistently, no increase was observed in the release of lactate dehydrogenase (LDH), a known marker of necrosis (Figure 2B), nor in annexin V positivity, typically associated with apoptotic death (Figure 2C); the fraction of annexin V-positive cells was, indeed, below 2% of the whole population in all the cell cultures tested.

To better define the effects of cytomix on cell growth, we next addressed cell cycle progression with flow cytometry. As shown in Figure 3A, the exposure of WT cells to IFNγ + TNFα caused an increase in the percentage of cells in G0/G1 (from 57.6 to 74.1%), paralleled by a marked decrease in both S (from 27.7 to 14.2%) and G2/M phases (from 9.6 to 4.1). Under the same condition, the changes in IRF1 KO cells were much less pronounced for both G0/G1 (from 56,7 to 60.8) and S phases (from 25.9 to 19.9%), while the percentage of G2/M cells decreased to the same extent as in WT cells, from 11.8 to 5.8%. Also, in line with these findings, the pattern of expression of proteins associated with the cell cycle checkpoints was compatible with an arrest in G0/G1 arrest (Figure 3B). In WT cells, indeed, the expression of the cell cycle inhibitor p21 transiently increased after 4 and 8 h of exposure to the cytomix. In parallel, the phosphorylation of Rb progressively decreased, with an almost complete dephosphorylation after 24 h, and the amount of cyclin D3 progressively lowered, disappearing at 24 h. All these changes were much less evident in IRF1 KO cells, where only a mild increase in p21 protein and a slight decrease in Rb phosphorylation were observed after 8 and 24 h of incubation with cytomix, respectively. Overall, these results support the hypothesis of a role for IRF1 transcription factor in the cytostatic effects observed. Consistently, cytokine-dependent cell cycle arrest in WT cells was paralleled by a massive and transient induction of IRF1 expression; as expected, no change was observed in IRF1 KO cells, where the protein remained undetectable at any incubation time.

The occurrence of senescence in cells undergoing cytokine-induced growth arrest was, then, investigated. To this end, we performed a senescence-associated β-galactosidase assay in WT and IRF1 KO cells incubated for 72 h in the presence of IFNγ + TNFα, along with the measurement of cytokine release and ICAM-1 expression, as markers of SASP (senescence-associated secretory phenotype). Results, presented in Figure 4, show no positivity to β-gal in neither cell models (Panel A). However, the incubation with cytomix caused a huge secretion of IL-6 and IP-10 in WT cells; the release of these mediators was impressively lower in IRF1 KO cells. IL-8 was, instead, strongly and equally released by both cell models. Under the same experimental condition, ICAM-1, undetectable in control untreated cells, was progressively induced in both cell models; this induction, however, was impressive in WT cells, while much less pronounced in IRF1 KO (Panel C), further supporting a role for IRF1 in the effects observed.

Since this transcription factor is a downstream target gene of STAT1 in the JAK/STAT signaling pathway, the molecular mechanisms underneath cytomix-dependent cell growth arrest were investigated by employing the JAK1/2 inhibitor baricitinib (Figure 5). To this concern, we have also recently demonstrated that the exposure of A549 cells to conditioned medium from spike S1-activated human macrophages triggers IRF1 expression through the JAK1/2–STATs axis [33,34]. Results shown in Panel A demonstrated that the exposure to IFNγ + TNFα caused the phosphorylation of STAT1 in both WT and IRF1 KO cells, an effect completely prevented, as expected, by baricitinib. The same drug also almost completely suppressed the cytokine-driven induction of IRF1 in WT cells, as well as the increase in p21 protein and the dephosphorylation of Rb in the same cells. Consistently, baricitinib almost completely hindered the decrease in cell viability induced by the treatment of WT cells with IFNγ + TNFα for 72 h; conversely, the modest decrease observed in IRF1 KO cells was not affected by the drug (Panel B). The inhibitor also prevented the induction of IL-6, IP-10 and ICAM-1 expression, but not that of IL-8 (Panel C).

## 3. Discussion

In a previous study we demonstrated that the exposure of A549 cells to conditioned medium (CM) obtained from SARS-CoV-2-activated human macrophages caused proliferative arrest [33]; by analyzing the cytokine content of this conditioned medium, we concluded that the IFNγ–IRF1 axis likely plays a central role in the induction of this cytostatic effect [34]. Based on these findings, the present study aimed to deepen this issue by assessing the contribution of single and combined inflammatory cytokines IFNγ, TNFα, IL-1β and IL-6 on growth arrest and on the secretory phenotype of A549 alveolar epithelial cells; in addition, we clarified the role of IRF1 by taking advantage of A549 cells knockout for this transcription factor. We are aware that the cell model adopted, as a cell line, does not properly reflect the conditions of the normal epithelium in vivo; however, A549 cells, although not primary cells, are considered today as a good model for type 2 alveolar epithelial cells.

Results obtained demonstrate that, although IFNγ alone actually caused a marked inhibition of cell proliferation in WT cells, the maximal effect was obtained with the simultaneous presence of IFNγ + TNFα in the culture medium. The same cytomix also induced profound alterations in cellular morphology with cells showing an elongated shape; no sign of cytotoxicity or apoptosis was, however, appreciable and no biochemical marker of cell death was induced. It is conceivable that more prolonged incubation times could result in a more pronounced damage phenotype; however, addressing a chronic effect of cytokines for longer times is not feasible in our cell model because of the high cell proliferation rate. Actually, our findings are in contrast with the literature data that describe a synergism of IFNγ plus TNFα as a trigger for apoptosis and pyroptosis in BEAS-2B airway epithelial cells [35]. Similarly, the same combination of cytokines has been described as an effective approach to induce cell death in cancer cells [36], and high levels of TNFα and IFNγ are supposed to be responsible for the inflammatory cell death in severe COVID-19 patients [28,29].

On the other hand, cytokines are notoriously described as able to induce cellular senescence as a response program to different stressors [37] and it is nowadays recognized that IFNγ plus TNFα are responsible for the induction of senescence in cancer cells, so that the use of these cytokines is proposed as a general mechanism for arresting cancer progression through the so-called cytokine induced senescence (CIS) [31,38]. Key features of senescent cells include decreased cell proliferation, increased activity of senescence-associated β-galactosidase (SA-β-gal) and the high expression of cyclin-dependent kinase inhibitor genes, such as p53, p21 and p16. Another remarkable hallmark of senescent cells is the acquisition of the so-called “senescence-associated secretory phenotype (SASP)”, which is characterized by the release of soluble factors in the surrounding environment, including proinflammatory cytokines and chemokines, matrix degrading enzymes and growth factors [32,39,40]. Moreover, the induction/secretion of adhesion molecules, such as ICAM-1 and VCAM-1, is nowadays recognized as a typical feature of senescent cells [39,41]. Although we did not observe any positivity to β-galactosidase expression (at least until 72 h of treatment), all other findings strongly sustain the induction of a senescent phenotype under our experimental conditions. Upon incubation with cytomix, indeed, A549 WT cells underwent a proliferative arrest associated with the up-regulation of p21 protein and the dephosphorylation of Rb, as well as to the secretion of pro-inflammatory cytokines and chemokines, such as IL-6, IL-8 and IP-10, and to a marked induction of ICAM-1. Interestingly, we have recently described a similar secretory phenotype for A549 cells upon incubation with CM isolated from spike S1-treated macrophages [42]. Based on our present findings, we can now hypothesize that those events were related to the acquisition of SASP by treated alveolar epithelial cells, in line with the literature evidence that proposes senescence-governed immune escalation upon SARS-CoV-2 infection as one of the factors leading to severe COVID-19 disease [43].

In order to define the molecular mechanisms responsible for the effects of cytomix, we then mostly focused on IRF1 transcription factor, that we have recently proposed as central to the immune-mediated effects of SARS-CoV-2 spike protein in A549 [42]. Results obtained in IRF1 KO cells demonstrate here that the observed effects of IFNγ + TNFα were largely IRF1-dependent, since all changes induced by cytomix were much smaller in IRF1-deficient than in WT cells. Cell growth inhibition in IRF1 KO cells was, indeed, less pronounced, with no evident modification of cell morphology. Moreover, the weak or even absent induction of SASP factors IL-6, IP-10 and ICAM-1 in IRF1 KO cells demonstrates that also the expression of these protein is under IRF1 control.

Lastly, by employing the JAK1/2 inhibitor baricitinib, we concluded that the molecular pathway responsible for IFNγ+TNFα-mediated effects likely involves the JAK/STAT/IRF1 axis. Baricitinib, initially approved for the treatment of rheumatic arthritis [44], is now employed for the treatment of hospitalized patients with severe COVID-19 [45], since one of the molecular mechanisms central to the development of the cytokine storm in COVID-19 is the JAK/STAT pathway [46,47]. Actually, by restraining the immune dysregulation associated with the severe form of the disease, the treatment with baricitinib effectively reduces the progression to mechanical ventilation or mortality. In particular, the drug proved effective in reducing the plasma concentrations of several pro-inflammatory cytokines, including IL-6, IL-1β and TNFα, in inducing the recovery of circulating lymphocytes and increasing the production of antibodies against spike protein [48,49,50]. In line with these findings, we show here that the drug, by suppressing STAT1 and IRF1 expression, completely prevented IL-6 and IP-10 release, as well as ICAM-1 induction, and protected cells from the decrease in cell viability. A similar picture has been recently described in human endothelial cells by Kandhaya-Pillai that, by targeting STAT1 with the JAK inhibitor ruxolitinib, proposed TNFα/IFNγ–STAT1 axis as the link between cytokine-mediated senescence and senescence-mediated cytokines that reinforce each other, amplifying the senescence-inflammation loop [51]. In addition, some recent in vitro studies on fibroblasts from Hutchinson–Gilford progeria syndrome (HGPS) patients revealed the beneficial effect of this drug which ameliorates HGPS cellular homeostasis and delays senescence [52,53,54]. Moreover, Gu et al. recently demonstrated the efficacy of baricitinib in the attenuation of pulmonary fibrosis in mice through the suppression of TGF-β1-mediated fibroblast activation and epithelial injury [55].

Interestingly, the production of IL-8 in response to cytomix was comparable in WT and IRF1 KO cells, demonstrating that the synthesis of this chemokine, already demonstrated to be STAT-independent [42], is also IRF1-independent. Consistently, the incubation with the JAK1/2 inhibitor baricitinib did not prevent IL-8 induction. This latter finding is consistent with evidence by Bronte and colleagues that showed no differences in IL-8 plasma levels of COVID-19 patients treated with baricitinib, despite a significant reduction in IL-1β, IL-6 and TNFα concentrations [48]. This result appears of particular relevance when considering the central role of the chemokine in the pathogenesis of ARDS [56], since it suggests that alternative therapeutic strategies have to be developed to target the neutrophil–IL-8 axis in severe COVID-19.

## 4. Conclusions

Our results demonstrate that the axis JAK/STAT/IRF1 is an upstream regulator of the cytokine-induced proliferative arrest and secretory phenotype in alveolar epithelial cells. Overall, these findings can open new field of investigations in the potential role of senescence-associated inflammation in pulmonary lung diseases; in this context, baricitinib could prove to be a promising therapeutic strategy for the attenuation of senescence-associated inflammation.

## 5. Materials and Methods

### 5.1. Cell Models and Experimental Treatments

Human wild-type A549 cell line (A549 WT, ab255450) and human IRF1 knockout A549 cell line (IRF1 KO, ab267042) were obtained from Abcam (Prodotti Gianni S.r.l., Milano, Italy) and cultured in RPMI1640 medium supplemented with 10% FBS and 1% penicillin/streptomycin at 37 °C in a humidified atmosphere with 5% CO_2_. For experiments, cells were seeded at a density of 2.5 × 10^4^ cells/cm^2^ in multi-well culture plates; after 24 h, cells were incubated in RPMI1640 with the required concentrations of IFNγ, IL-1β, TNFα and IL-6 (R&D Systems by Biotechne, Milano, Italy), alone or combined. Where indicated, cells were pre-treated with 1 µM baricitinib for 1 h before the addition of the cytokines; the inhibitor was left in the culture medium throughout the experiment.

### 5.2. Cell Viability

A549 cells, seeded in 48-well plates, were treated as required by the experimental plan for 24, 48 and 72 h. Cell viability was evaluated with the resazurin assay [57], based on the reduction in the non-fluorescent compound resazurin by viable cells into the fluorescent resorufin, that accumulates into the medium. After the treatments, cells were, thus, incubated for 1 h at 37 °C with fresh growth medium supplemented with 44 μM resazurin, and fluorescence was then measured at 572 nm with a fluorimeter (EnSpire Multimode Plate Readers; PerkinElmer, Monza, Italy). Cell viability was calculated after subtraction of cell-free background, and expressed as arbitrary fluorescence unit

### 5.3. Cell Death

To evaluate cell death, cells were seeded in 24-well plates and treated as required. After 72 h, the release of lactate dehydrogenase (LDH) was determined in the extracellular medium with the Lactate Dehydrogenase Activity Assay kit (Merck, Milano, Italy), according to the manufacturer’s instructions. The absorbance values at 490 nm were measured with a microplate reader (EnSpire Multimode Plate Readers; PerkinElmer). A positive control was prepared by exposing cells to 1 µM staurosporine for 24 h. Data are expressed as a percent of the absorbance measured in control untreated WT cells.

Apoptosis was evaluated with a Annexin V FITC/Dead Cell Apoptosis Kit for flow cytometry (Thermo Fisher Scientific, Monza, Italy). After a treatment for 24 h with IFNγ + TNFα, cells were harvested, washed twice with PBS and resuspended in 200 μL binding buffer supplemented with 10 μL of Annexin-V FITC and 5 μL of propidium iodide (PI). After an incubation for 20 min at RT in the dark, 300 μL of binding buffer was added, and a flow cytometric analysis was performed using a Cytoflex flow cytometer (Beckman Coulter, Milano, Italy).

### 5.4. Cell Cycle Analysis

For the analysis of the cell cycle, A549 cells were seeded in 12-well plates. The day after, cells were treated with cytokines for 24 h, then harvested through trypsinization and incubated for 18 h at 4 °C in a hypotonic solution containing 50 μg/mL propidium iodide (PI) and 10 μg/mL RNaseA. Cell cycle distribution was then determined using a Cytoflex flow cytometer (Beckman Coulter).

### 5.5. Western Blot Analysis

Cell lysates obtained with LDS sample buffer (Thermo Fisher Scientific, Monza, Italy) were employed for the analysis of protein expression as already described [33]. 20 µg of proteins were separated on Bolt™ 4–12% Bis-Tris mini protein gel (Thermo Fisher Scientific) and transferred to PVDF membranes (Immobilon-P membrane, Thermo Fisher Scientific). Membranes were incubated for 1 h at room temperature (RT) in blocking solution (4% non-fat dried milk in TBST, Tris-buffered saline solution + 0.5% Tween), then overnight at 4 °C with anti-IRF1, anti-p21^WAF^, anti-pRB-S780, anti-pSTAT1-Tyr701, or anti-ICAM-1 rabbit polyclonal antibody or anti-cyclin D3 monoclonal antibody (1:2000, Cell Signaling Technology, Euroclone, Pero (MI), Italy) in TBST containing 5% BSA. Anti-vinculin mouse monoclonal antibody (1:2000, Merck) was used as a loading control. Horseradish peroxidase (HRP)-conjugated secondary antibodies (anti-rabbit and anti-mouse IgG, Cell Signaling Technology) were employed (1:10,000) and immunoreactivity was visualized with SuperSignal™ West Pico PLUS Chemiluminescent HRP Substrate (Thermo Fisher Scientific). Western blot images were captured with an iBright FL1500 Imaging System (Thermo Fisher Scientific) and analyzed with iBright Analysis Software (version 5.0).

### 5.6. Senescence-Associated β-galactosidase (SA-β-gal) Staining

SA-β-galactosidase activity was measured by using Senescence beta-galactosidase Staining kit (Cell Signaling Technology) following the manufacturer’s instructions. Briefly, A549 cells, treated for 72 h with IFNγ + TNFα, were washed with PBS and fixed for 15 min at RT. Then, after two washes with PBS, cells were incubated for 24 h at 37 °C without CO_2_ in freshly prepared staining solution. Cells were observed under microscope for image acquisition.

### 5.7. Cytokine Analysis

For the measurement of cytokine release, cells were seeded in 24-well plates and treated as required. After 72 h, the incubation medium was collected and analyzed for cytokine content with Quantikine ELISA Kits for IL-6, IL-8 and CXCL10/IP-10 (R&D Systems by Biotechne) according to the manufacturer’s instructions. Cytokine levels in the extracellular medium (pg/mL) are expressed relatively to the cell number in each well.

### 5.8. RT-qPCR Analysis

For the analysis of gene expression, cells were seeded in 24-well plates and treated as required. After 24 h, total RNA was isolated with the GeneJET RNA Purification Kit (Thermo Fisher Scientific) and reverse transcribed with the RevertAid First Strand cDNA Synthesis Kit (Thermo Fisher Scientific). qPCR was then performed on a StepOnePlus Re-al-Time PCR System (Thermo Fisher Scientific) by employing specific forward/reverse primer pairs (Table 1) and SYBR™ Green Master Mix (Thermo Fisher Scientific). The amount of the genes of interest w calculated with the 2^∆Ct^ method [58] after normalization for the expression of the housekeeping gene RPL15.

### 5.9. Statistical Analysis

GraphPad Prism 9 (GraphPad Software, CA, USA) was used for statistical analysis. *p* values were calculated with ordinary One-way ANOVA for multiple comparisons or Student’s t test, as specified in the legend of each Figure. *p*-values < 0.05 were considered statistically significant.

### 5.10. Materials

R&D Systems was the source of recombinant human cytokines: HEK293-expressed IFNγ; E. coli-derived IL-1β/IL-1F2 protein; HEK293 expressed TNFα; and HEC293-expressed IL-6. Endotoxin-free fetal bovine serum was purchased from Thermo Fischer Scientific, while baricitinib (Cayman Chemicals, Ann Arbor, MI, USA) was from Vinci-Biochem S.r.l., Firenze, Italy. Merck (Milano, Italy) was the source of all the other chemicals, unless otherwise specified.

## Figures and Tables

**Figure 1 ijms-25-03463-f001:**
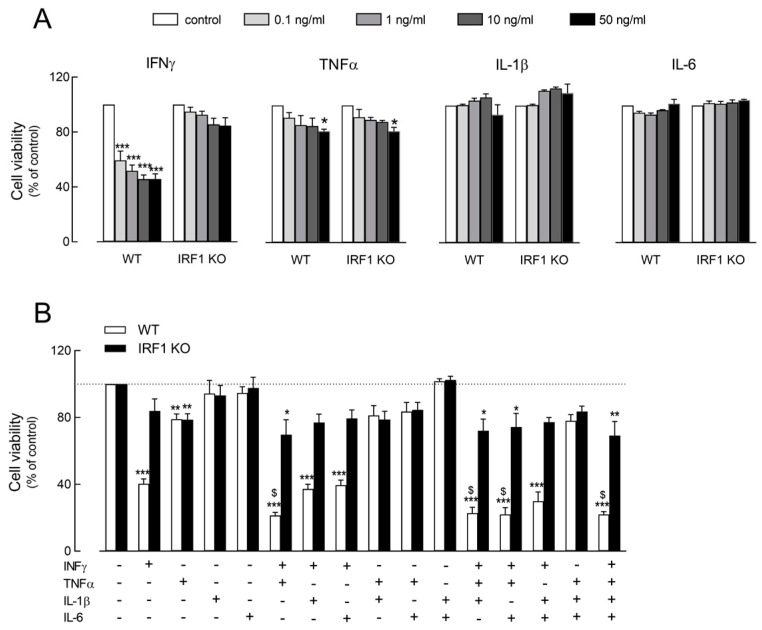
Effect of cytokines on the viability of A549 WT and IRF1 KO alveolar epithelial cells. Panel (**A**): Cells were incubated in RPMI1640 medium in the absence (control) or in the presence of the indicated concentrations of INFγ, TNFα, IL-1β and IL-6. After 72 h, cell viability was assessed through resazurin assay, as described in Methods. Each point, calculated as a percent of the control, is the mean ± SD of three determinations in a representative experiment that, repeated three times, gave comparable results. * *p* < 0.05, *** *p* < 0.001 vs. control with One-way ANOVA. Panel (**B**): Cells were incubated for 72 h in the presence of the indicated cytokines (10 ng/mL IFNγ; 50 ng/mL IL-1β, TNFα and IL-6) and cell viability was calculated as a percent of the control untreated cells (dotted line). Bars are means ± SEM of three experiments, each performed in triplicate. * *p* < 0.05, ** *p* < 0.01, *** *p* < 0.001 vs. control; ^$^ *p* < 0.05 vs. IFNγ with One-way ANOVA.

**Figure 2 ijms-25-03463-f002:**
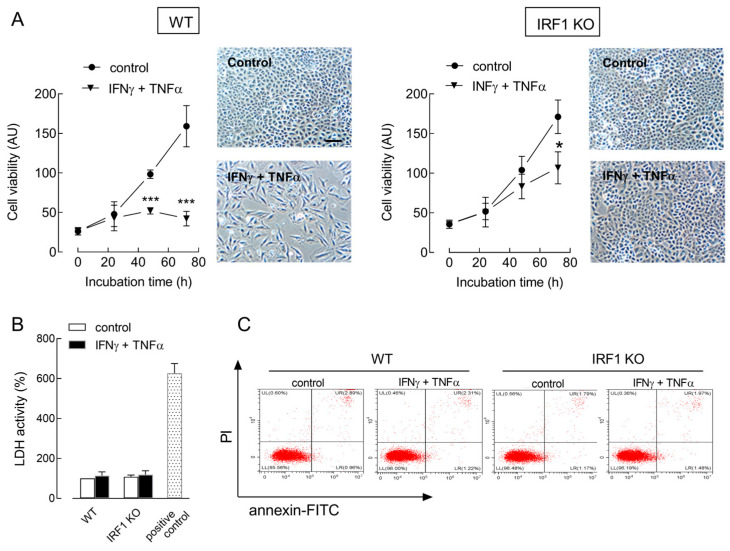
Effect of cytomix on the proliferation of A549 WT and IRF1 KO cells. Cells were incubated in the absence (control) or in the presence of 10 ng/mL INFγ + 50 ng/mL TNFα (cytomix). Panel (**A**): At the indicated times, cell viability was determined with the resazurin assay, as described in Methods. * *p* < 0.05, *** *p* < 0.001 vs. control with Student’s *t*-test. Representative images of cell cultures treated for 72 h are shown; bar = 100 µM. Panel (**B**): LDH activity was determined after 72 h as described in Methods. Data are expressed as a percent of the control cell activity. A positive control of necrotic cells, obtained with a 24 h exposure to 1 µM staurosporine, is shown. Data are the mean ± SD of four determinations in a representative experiment, which, repeated three times, gave comparable results. Panel (**C**): After 24 h, apoptotic cell death was evaluated with flow cytometry through Annexin V-FITC/Propidium iodide staining, as detailed in Methods. Flow cytometry graphs in a representative experiment are shown. The experiment was repeated three times with comparable results.

**Figure 3 ijms-25-03463-f003:**
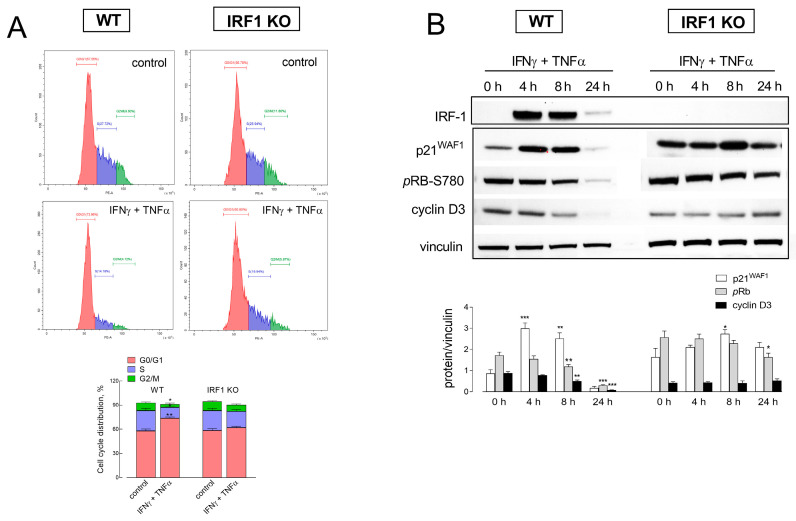
Effect of cytomix on cell cycle progression in A549 WT and IRF1 KO cells. Cells were incubated in the absence (control) or in the presence of 10 ng/mL IFNγ + 50 ng/mL TNFα. Panel (**A**): After 24 h of incubation, cells were harvested, stained with propidium iodide and analyzed for cell cycle with flow cytometry, as detailed in Methods. Plots obtained in a representative experiment are shown. Cell distribution in each phase of the cell cycle is also shown (bottom); bars are the mean ± SEM of data obtained in three independent experiments. * *p* < 0.05, ** *p* < 0.01 vs. control with Student’s *t*-test. Panel (**B**): At the indicated times, the expression of the indicated proteins was assessed by means of Western Blot analysis, as detailed in Methods. Representative blots are shown along with the mean ± SEM of the densitometric analysis of three different experiments. * *p* < 0.05, ** *p* < 0.01, *** *p* < 0.001 vs. 0 h (untreated cells) with One-way ANOVA.

**Figure 4 ijms-25-03463-f004:**
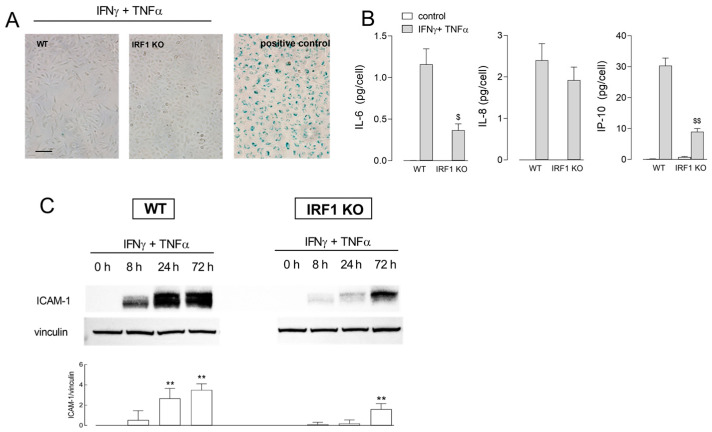
Effect of INFγ + TNFα on the induction of a senescent phenotype in A549 WT and IRF1 KO. Cells were incubated for 72 h in the absence (control) or in the presence of 10 ng/mL INFγ + 50 ng/mL TNFα. Panel (**A**): Cellular senescence was assessed with β-galactosidase staining. Representative images are shown with a positive control represented by endothelial cells (HUVECs) maintained in culture for 10 passages. Panel (**B**): The amount of the indicated cytokines was measured with an ELISA assay, as described in Methods. Data are means ± SEM of three independent experiments, each performed in triplicate. ^$^ *p* < 0.05, ^$$^ *p* < 0.01 vs. WT cells with Student’s *t*-test. Panel (**C**): ICAM-1 protein was assessed by means of Western Blot analysis at the indicated times of treatment, as detailed in Methods. Representative blots are shown along with mean ± SEM of the densitometric analysis of three different experiments. ** *p* < 0.01 vs. 0 h (untreated cells) with One way ANOVA.

**Figure 5 ijms-25-03463-f005:**
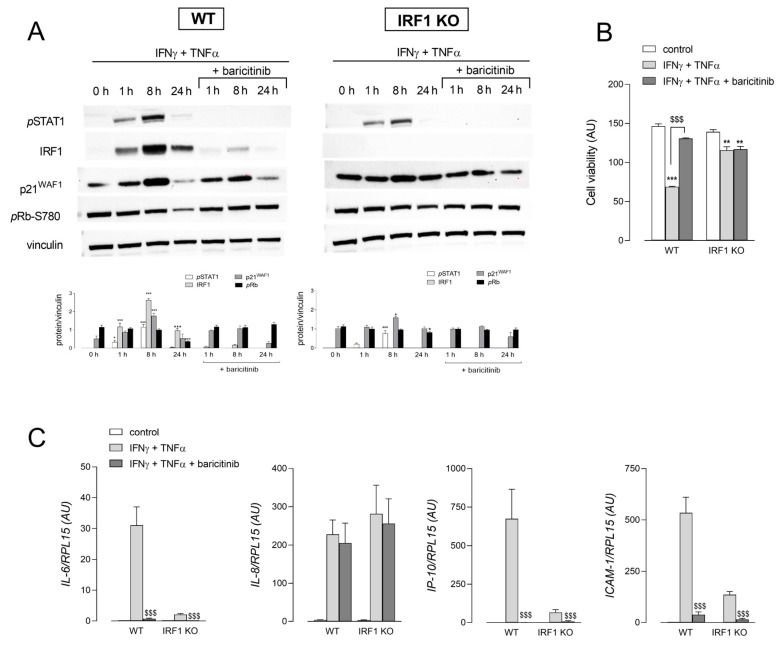
Effect of baricitinib on the cytomix-induced expression of senescence markers. Panel (**A**): The cells were incubated in the absence (0 h) or in the presence of 10 ng/mL INFγ + 50 ng/mL TNFα for the indicated times; 1 µM baricitinib was added 1 h before the treatment and maintained throughout the incubation. The expression of the indicated proteins was assessed by means of Western Blot analysis, as detailed in Methods. Representative blots are shown along with mean ± SEM of the densitometric analysis of three different experiments. * *p* < 0.05, *** *p* < 0.001 vs. 0 h (untreated cells) with One-way ANOVA. Panel (**B**): After 72 h of treatment, cell viability was assessed through resazurin assay, as described in Methods. Each point is the mean ± SD of four determinations in a representative experiment that, repeated three times, gave comparable results. ** *p* < 0.01, *** *p* < 0.001 vs. control; ^$$$^ *p* < 0.001 vs. INFγ + TNFα with Student’s *t*-test. Panel (**C**): The expression of the indicated genes was measured after 24 h of incubation under the indicated conditions by means of RT-qPCR, as described in Methods. Bars are means ± SEM of three independent experiments, each performed in duplicate. ^$$$^ *p* < 0.001 vs. INFγ + TNFα with Student’s *t*-test.

**Table 1 ijms-25-03463-t001:** Sequences of the primer pairs employed for RT-qPCR analysis.

Gene/Protein Name (Gene ID)	Forward Primer	Reverse Primer
*IL-6/*IL6 (3569)	AACCTGAACCTTCCAAAGATGG	TCTGGCTTGTTCCTCACTACT
*CXCL8/*IL-8 (3576)	ACTGAGAGTGATTGAGAGTGGAC	AACCCTCTGCACCCAGTTTTC
*ICAM1/*ICAM-1 (3383)	TGAACCCCACAGTCACCTATG	CTCGTCCTCTGCGGTCAC
*CXCL1010/*IP-10 (3627)	GTGGCATTCAAGGAGTACCTC	TGATGGCCTTCGATTCTGGAT

## Data Availability

Data are available in https://osf.io/gr9c6/ (accessed on 8 March 2024).

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
