# Peer review of "IRF1 Mediates Growth Arrest and the Induction of a Secretory Phenotype in Alveolar Epithelial Cells in Response to Inflammatory Cytokines IFNγ/TNFα"

_ijms, 2024, doi:10.3390/ijms25063463_

Round 1

Reviewer 1 Report

Comments and Suggestions for Authors

1. Brief Summary:

The paper investigates the effects of inflammatory cytokines on cell viability in A549 wild-type (WT) and IRF1 knockout (KO) cells. The study demonstrates differential responses to IFNγ, TNFα, IL-1β, and IL-6 in the two cell types, highlighting the role of IRF1 in mediating cytokine-induced growth arrest. The main strengths of the paper lie in its clear experimental design, detailed results presentation, and the identification of potential therapeutic targets for inflammatory lung diseases.

2. General Concept Comments:

2a. Article: The paper effectively elucidates the impact of cytokines on cell viability in A549 cells. However, there are areas of weakness in the discussion of the mechanistic insights underlying the observed effects. Additionally, the methodological section could benefit from more detailed descriptions of experimental procedures and controls to enhance reproducibility.

2b. Review: The review topic is relevant and addresses a significant gap in knowledge regarding the role of IRF1 in cytokine-induced growth arrest. The references cited are appropriate and support the study's findings. However, further discussion on the clinical implications of the results and potential future research directions would enhance the manuscript's overall impact.

3. Special Comments:

The authors should consider providing more context on the clinical relevance of their findings, particularly in the context of inflammatory lung diseases. Additionally, discussing the limitations of the study and proposing avenues for further investigation would strengthen the paper. Overall, the paper presents valuable insights into the molecular mechanisms underlying cytokine-induced growth arrest, but further refinement in the discussion section and future perspectives would enhance its scientific impact.

Author Response

We would like to thank the Reviewer for the attention he/she paid to our manuscript and for his/her comments, that helped us to significantly improve the manuscript.

Regarding the first issue, we must apologize, because we realized that we had omitted from the final version of the manuscript the part concerning the explanation of the experimental treatments. Now we have added this part and modified the whole section to better explain procedures and controls.

Then, we have tried to better discuss our findings from a clinical perspective, as well as to explain the limitations of our study and to highlight possible future implications. As a result, the manuscript has been extensively revised and we believe it is more clear now.

Reviewer 2 Report

Comments and Suggestions for Authors

Paper by Luciani et al. highlights the role of inflammation in the development of senescence and senescence-associated secretory phenotype (SASP) in alveolar epithelial cells, both mechanisms involving IFNγ, TNFα cytokines and STAT and IRF1-dependent. Interestingly, this study is framed within of current lung diseases as COVID-19 and acute respiratory distress syndrome (ARDS). Paper is well written, easy to read and possess scientific robustness. However, Authors need to improve their manuscript, addressing some points.

1)        Authors should change title to a clearer one. The correlation between IFNγ/TNFα cytokines and IRF1 is not clear here.

2)        Authors should improve abstract, and, among diseases caused by cytokine storm, they should mention COVID-19, since it is extensively discussed in article. In addition, they should specify in the abstract which effect is exerted by baricitinib. Moreover, they should report results obtained in IRF1 KO cells in results section of abstract.

3)        Authors should provide the original Western image (uncropped) of three replicates for each studied target. In addition, they should report the densitometric analysis of Western results present in Figures 4C, 5A-B.

4)        Authors should explain acronyms when they appear for the first time in the text (i.e., PMN or BAL in lines 45-46).

5)        Authors should justify in the text why they used all cytokines at the concentration of 50 ng/mL except for IFNγ which was used at concentration of 10 ng/mL.

6)        Authors should clearly indicate the link between the IRF1 factor and STAT1 and why they used the JAK1/2 inhibitor baricitinib in their study, both in the Results and Discussion sections.

7)        Authors should add Conclusions and possible future challenges to their paper.

Comments on the Quality of English Language

Minor editing of English language are required.

Author Response

We would like to thank the Reviewer for his/her collaborative criticisms; here our response to his/her comments:

  1. According to his/her suggestion, we have modified the title of the manuscript, so as to better explain the link between cytokines and IRF1. However, if this change creates problems for the editorial process, we prefer to keep the original version.
  2. We have revised the Abstract, so as to cope with the Reviewer’s suggestions which were all actually relevant.
  3. All original Western Blot images are now provided, and the densitometric analyses have been added in Figures 4C, 5A and 5B, as requested.
  4. All acronyms are now explained, as requested.
  5. Cytokines concentration in cytomix was chosen based on results obtained in Panel A of Figure 1 (lines 111-112 of the original manuscript); for sake of clarity, we now specify that, for IFNγ, we chose the minimal concentration with the maximal effect.
  6. The link between JAKs, STATs and IRF is now better explained throughout the manuscript, as well as the reason why we employed baricitinib, as requested by the Reviewer.
  7. According to the suggestion of both Reviewers, we have better discussed our findings from a clinical perspective, as well as explained the limitations of our study and highlighted possible future implications. We hope that the revisions made throughout the Discussion enhance the scientific impact of our contribution.

Round 2

Reviewer 2 Report

Comments and Suggestions for Authors

Authors fully complied with the reviewer's requests.